# Informing the Design of Data Visualization Tools to Monitor the COVID-19 Pandemic in Portugal: A Web-Delphi Participatory Approach

**DOI:** 10.3390/ijerph191711012

**Published:** 2022-09-02

**Authors:** Ekaterina Ignatenko, Manuel Ribeiro, Mónica D. Oliveira

**Affiliations:** 1Centre for Management Studies of Instituto Superior Técnico (CEG-IST), Universidade de Lisboa, Av. Rovisco Pais, 1, 1049-001 Lisboa, Portugal; 2Centro de Recursos Naturais e Ambiente (CERENA), Instituto Superior Técnico, Universidade de Lisboa, Av. Rovisco Pais, 1, 1049-001 Lisboa, Portugal; 3iBB—Institute for Bioengineering and Biosciences and i4HB—Associate Laboratory Institute for Health and Bioeconomy, Instituto Superior Técnico, Universidade de Lisboa, Av. Rovisco Pais, 1, 1049-001 Lisboa, Portugal

**Keywords:** dashboard, COVID-19, data visualization tools, Delphi process, Portugal

## Abstract

Due to the large amount of data generated by new technologies and information systems in the health arena, health dashboards have become increasingly popular as data visualization tools which stimulate visual perception capabilities. Although the importance of involving users is recognized in dashboard design, a limited number of studies have combined participatory methods with visualization options. This study proposes a novel approach to inform the design of data visualization tools in the COVID-19 context. With the objective of understanding which visualization formats should be incorporated within dashboards for the COVID-19 pandemic, a specifically designed Web-Delphi process was developed to understand the preferences and views of the public in general regarding distinct data visualization formats. The design of the Delphi process aimed at considering not only the theory-based evidence regarding input data and visualization formats but also the perception of final users. The developed approach was implemented to select appropriate data visualization formats to present information commonly used in public web-based COVID-19 dashboards. Forty-seven individuals completed a two-round Web-Delphi process that was launched through a snowball approach. Most respondents were young and highly educated and expressed to prefer distinct visualisation formats for different types of indicators. The preferred visualization formats from the participants were used to build a redesigned version of the official DGS COVID-19 dashboard used in Portugal. This study provides insights into data visualization selection literature, as well as shows how a Delphi process can be implemented to assist the design of public health dashboards.

## 1. Introduction

Nowadays, due to the massive data generation in all areas, new challenges exist for the selection and visualization of relevant information and deriving value from it. Such challenges apply to healthcare in integrating and making available public health information [1]. As a consequence, new data visualization tools have been emerging, allowing the representation of data that stimulates human perceptual and cognitive abilities for problem-solving [2]. Since visual systems are powerful mechanisms to detect patterns and to represent large amounts of information, they can be crucial for facilitating data interpretation and monitoring and for better policy-making. Through data visualization, such as charts or graphs, the data are depicted in ways that allow viewers to experience them in a new light, exploring the unseen patterns and relationships within the data [3,4], being required for dashboards design meeting stakeholders’ and users’ needs in an effective way. Within this context, participatory methods have been adopted to improve the usability of interactive systems by collecting and analysing direct input from users [5,6]. The integration of stakeholders during the process ensures the acceptability of the developed solution since participatory methods combine the information from a diversity of sources more efficiently than quantitative or qualitative methods alone [7].

However, despite the highlighted importance of stakeholders’ opinions for dashboard design, the number of studies that integrate participatory methods to explore visualization options of dashboards is limited [8]. Most methods focus on determining the needs and requirements for dashboard design and functionalities rather than exploring the most preferred visualization format between suggested visual options [9,10,11,12,13].

In this paper, attention is given to the development of participatory methods to inform the selection of data visualization formats in the context of dashboard design for the COVID-19 pandemic. The research aims to provide a rationale for selecting appropriate visualizations, both considering theory-based evidence and the perspective of dashboard end-users.

In December 2019, a local outbreak of initially unknown respiratory illness was detected in Wuhan (Hubei, China) and was rapidly identified as being caused by coronavirus SARS-CoV-2. On 11 March 2020, coronavirus disease 2019 (COVID-19) was announced as a pandemic by the World Health Organization (WHO) [14], and in October 2021 (25 October 2021), 245 million confirmed cases and almost 5 million deaths had been registered, together with severe social, economic and health impacts [15,16].

Along the pandemic, different policies for the detection and containment of clusters of infection have been established to control the propagation of the virus. In such context, many data visualization tools have emerged for decision support—for instance, web-based dashboards have been developed to facilitate the transmission of relevant information to the general population and to promote an understanding of the COVID-19 data by the public [17]. Public web-based COVID-19 dashboards ultimately have shared a common objective: to serve as both a communication tool and a call for individual and collective action to respond to the COVID-19 pandemic [18]. In fact, as described by Dasgupta and Kapadia [19], “publicly available dashboards chronicling the COVID-19 pandemic have become ubiquitous, a staple of news outlets and health department communications” and the dashboard paradigm is here to stay in public health.

Most existing web-based public COVID-19 dashboards focus on epidemiological indicators, such as incidence (number of new COVID-19 cases during a specified time interval) and mortality (number of deaths caused by COVID-19 during a specified time interval). These indicators can be reported at different geographic levels, representing spatial distributions or trends over time, predominantly by day, to show the evolution of the pandemic and the effects of implemented policies [20,21]. In addition to geographic and temporal distributions, dashboards are used to analyse data by other characteristics, the most common being age and gender [18].

Visualization techniques have been front-and-centre in the efforts to communicate the science around COVID-19 and related public health information to the general population [22]. Public web-based COVID-19 dashboards make available visual information to transmit indicators usually in the form of graphs or charts, maps and tables [18]. However, overall underuse of known and proven delivery visual techniques is a common issue for COVID-19 dashboards, which can mislead both unintentionally and intentionally if the data visualizations are not accurately selected and represented [23], and users need to be involved to ensure that dashboards are comprehensible [24].

Multiple participatory methods such as interviews, surveys, workshops and Delphi processes have been used to understand and collect users’ views in health contexts [25]. In this study, we selected the Web-Delphi participatory process as it enables to involve a large and heterogeneous number of potential dashboard users that are located in different regions, the high integration and inclusion of different types of information [26], as well as it enables group interaction [27,28,29,30], with these being key features for gathering information to assist dashboard design in our context. Moreover, the Delphi approach facilitates convergence between participants by constant learning during the process and flexibility for participants to change their views in light of all participants’ views [31]. It is also noteworthy that, to our knowledge, we did not find studies reporting transparently the use of the Delphi method for selecting data visualization formats, despite the promising opportunity of integrating this approach in this context [32].

This research has been developed within the Spatial Data Sciences for COVID-19 Pandemic (SCOPE) project. The SCOPE project aims to develop a functional software prototype for spatial risk management, providing daily updates of health indicators maps related to the COVID-19 pandemic. This study contributes to one of the project objectives: to bridge the gap between the creation of risk maps and dashboards related to the pandemic, and the use of such maps to support decision-making and the design of policies. Specifically, a Web-Delphi process was developed to understand the preferences of the general public regarding distinct data visualization formats of COVID-19-related information.

The remainder of this paper is structured as follows. Section 2 is organized around the development and implementation of a novel approach—based on a Web-Delphi process—to assist in the selection of the most appropriate data visualization formats for public web-based COVID-19 dashboards. The results of this implementation are presented in Section 3, including an illustrative development of the Portuguese DGS COVID-19 Dashboard that considers the results of the Web-Delphi process. Section 4 reflects upon the case study results, and Section 5 provides key remarks about the developed research.

## 2. Materials and Methods

A specifically designed approach, based on a Delphi process, was developed to understand the preferences and views of the general public regarding distinct data visualization formats to be used in a dashboard for the COVID-19 pandemic—see Figure 1. The first step of the approach was to select indicators commonly used in public web-based COVID-19 dashboards. Then a set of appropriate data visualization formats for presenting those indicators were chosen considering the theory-based evidence reported in the literature and the data visualization formats that are easy to implement in dashboard platforms. Finally, a modified Web-Delphi process was designed and implemented to acquire collective knowledge about data visualization format preferences for each indicator from the pre-selected set. After having gone through two rounds, the Web-Delphi results were analysed by the level of agreement and could be further analysed by a smaller group of participants. With the results of the Web-Delphi, we illustrate how the web-based COVID-19 dashboard provided by the Portuguese national health authority could be adjusted to consider the views of end-users. Underlying the choice of a Web-Delphi participatory process are the advantages of potentiating group interaction, not requiring face-to-face contact, removing geographical barriers, and allowing the involvement of a large number of experts [27].

### 2.1. Indicators’ Selection

The selection of appropriate and well-designed indicators to integrate dashboards is vital for dashboard effectiveness and performance, deserving specific attention. Since the focus of this study is to explore which visualization formats are preferred—according to views of the public—to display dashboard information, the starting point was to choose a set of indicators commonly used for web-based public COVID-19 dashboards. For this purpose, we used the results from the study “Features Constituting Actionable COVID-19 Dashboards: Descriptive Assessment and Expert Appraisal of 158 Public Web-Based COVID-19 Dashboards” [18]—which explores and reviews the characteristics of 158 public web-based COVID-19 dashboards by analysing their features, namely the key performance indicators and their frequency, and which summarizes the types of data provided.

Accordingly, for use within the Web-Delphi process, we selected the following set of indicators related to COVID-19 incidence, stratified by age, gender and region, as these are the most frequently used indicators in public COVID-19 dashboards:Daily number of new confirmed cases.Total number of confirmed cases.Daily number of new confirmed cases per region.Total number of confirmed cases by age group.Total number of confirmed cases by gender.Total number of confirmed cases by region.Total number of confirmed cases by gender and age group.

### 2.2. Selection of Pre-Set of Data Visualization Formats

This step used a combination of the data visualization processes of Dastani [33] with Kirk’s approach [3], i.e., identified the most appropriate data visualization formats for a certain indicator, considering the data structure and communication purpose. First, according to the Dastani process [33], the variables of each indicator are identified and classified according to their attribute types. Variables’ attribute types can be quantitative or categorical (qualitative), which contain numerical values or nominal/ordinal values (e.g., geographic regions, age groups), respectively. For the sake of simplicity, one can use the designation used by Helfman [34], where the variable’s attribute types used in data visualization formats can be represented by a string where its length corresponds to the number of variables, and the letter to the variable’s attribute type, categorical (C) or quantitative (Q) (e.g., “CQ” means that this data visualization format uses two variables, 1 categorical and 1 quantitative).

Then, using the Kirk’s method’s taxonomy [3], one reflects upon the communication purpose that the data visualization format intends to transmit, this way reducing the range of suitable chart types within each method family by accounting for the nature of the variables in question.

To implement these procedures, the decision table presented in Table 1 was developed, using information from the literature to enumerate the most common data visualization formats and the correspondent variable’s attribute types that are appropriate for the specific communication purpose. This table uses Kirk’s categorization of data visualization formats according to the specific communication purpose, and it combines knowledge about the quantity and type of typical data variables normally used with these visualizations [3]. It allows predefining the set of most appropriate data visualization formats for a certain indicator, not providing a unique answer and requiring further reflection related to the indicators’ and dashboard context. 

Afterwards, the following procedure was adopted:The data structure of the previously selected indicators was identified, i.e., the number of variables and their attribute types determined.The communication purpose of data visualization formats for each indicator was identified, using the categorization provided by Kirk.The set of data visualization formats was defined by mapping the previously determined data structure and communication purpose with a developed decision table.

This procedure led to the selection of data visualization formats to consider in the Delphi processes and displayed in Table 2.

Additionally, choropleth, bubble, dasymetric and point maps [35] were selected to represent indicators which entailed a geographic location.

Since the adopted process offers a set of alternative data visualization formats for a certain dataset rather than identifying the most adequate visualization, the involvement of potential dashboard users was performed through a Web-Delphi participatory process.

### 2.3. Delphi Process

The Delphi is a structured group communication process which uses a number of questionnaires or rounds with controlled feedback to collect and deliver information with the objective to promote or achieve a group consensus [36]. Delphi processes have been used to identify, forecast, and investigate group attitudes, needs, and priorities through a series of rounds in which participants’ viewpoints are gathered through their individual responses to the same questionnaire. Thus, while maintaining anonymity, a summary of the responses is provided back to the participants, who may review their answers in the following rounds as a result of this collective knowledge [27,36,37].

The main objective of the proposed Web-Delphi process was to identify the most preferred visualization formats by users, so as to inform the design of dashboards that are used by the public to monitor the evolution of the COVID-19 pandemic in Portugal. The starting point was to use the list of indicators and indicators’ formats selected in Section 2.1 and Section 2.2 within a Web-Delphi process to obtain the preferences from potential dashboard users.

The proposed Delphi process is a modified one since the credibility of the questions elaborated for consequent rounds is ensured by the scientific background provided in the previous steps [38]. The study population were Portuguese citizens, users of public social media networks that accepted to participate in the Web-Delphi process. Participants were defined as a sample of possible dashboard users who were recruited according to the ‘snowballing’ method [39] using social media. The snowball sampling procedure consisted in posting an invitation to the researchers’ public social media networks, not only to participate in the Delphi process but also to ask network users to forward the invitation (reposting) to other users to participate in an effort to raise and widen the number of participants. Initially, invitations were posted on researchers’ social media networks and announced using hashtags: #coronavirus, #covid and #covid19. The invitation post included a brief presentation of the research and of the Delphi process and a link to the Web-Delphi questionnaire. Therefore, a hyperlink to the first round of the Delphi process was created and shared on 15 June 2021 using Facebook, Instagram and LinkedIn to capture the attention of a high number of individuals with different characteristics and backgrounds. This method for recruiting respondents was not intended to generate a representative sample of the study population but to constitute a sample of individuals with divergent opinions, thereby representing a wide spectrum of points of view. The diversity of these views can then be indirectly confronted with each other by the Delphi process.

As attrition is likely to increase with each round, to avoid participant fatigue and to guarantee meaningful results, two rounds were selected as an a priori feature of the Web-Delphi process. The first round took the form of a structured questionnaire with a total of 12 questions, including statements making use of the information displayed in Table 2, with the objective of acquiring collective knowledge about which data visualization format was preferred for each indicator from the pre-selected set. The elaborated questionnaire consisted of four main parts. In the first part, generic and clear information about the Delphi process was provided, so as enable an understanding and to remove ambiguity; and a consent agreement form had to be read and filled by the participant, conveying anonymity and voluntary participation in the study. Each participant was asked about socio-demographic information regarding age group, gender, and level of education. Lastly, participants were asked if they have working experience in health and familiarity with COVID-19 dashboards.

The second part explored the data visualization formats of the selected temporal COVID-19 indicators, namely the daily and total number of new confirmed cases. Participants were invited to analyse different time intervals (one week versus three months) in alternative data visualization formats that enable to understand variability and trends in COVID-19 cases at different temporal resolutions. Figure 2 displays a snapshot of the Web-Delphi platform, showing the first question with available options for the indicator “Daily number of new confirmed COVID-19 cases”. 

The third part explored the data visualization formats corresponding to the total number of confirmed cases since the beginning of the COVID-19 pandemic, aggregated by region, age group and/or gender.

The last part of the questionnaire included in the first question the set of map visualizations regarding the indicator of COVID-19 incidence, including choropleth, bubble, dasymetric and point maps. Another question aimed to understand the participants’ preference regarding the map format for portraying the spatial distribution of COVID-19 infection risk. Therefore, two map formats were suggested: the first one corresponded to the map with constant infection risk within the administrative unit, and another represented the map that portrayed the infection risk continuous in the space (a map specifically developed within the scope of the SCOPE research project).

The questionnaire gave participants the opportunity to provide further comments regarding each question, which could write a free-text comment at the end of the questionnaire. Several screens from the first-round questionnaire are available as Appendix A. In the final section of the questionnaire, the participants were asked to leave their e-mail in order to be contacted for the second round—only with that information participants could be invited to participate in that stage.

After the first Delphi round, individual participants’ answers were synthesized, and a statistical summary of percentage votes for each item was calculated and fed into round 2. In round 2 participants were given their own responses and a synthesis of all respondents’ votes for the distinct data visualization formats—participants were given the opportunity to confirm or revise their answers, considering the group information provided, within a collective learning task. The participants could also visualize the comments of the other respondents from round 1.

SurveyHero [40] was selected to implement the Delphi process, due to the possibility to integrate images within a multiple-choice option and due to the functionality of users zooming the images and selecting all the details. 

Although the Web-Delphi process can then be complemented with a workshop in which a smaller group of potential dashboard users can discuss and take insights from the results of the Delphi process, following a participatory knowledge construction process [27], this additional task was not performed in this study [27].

### 2.4. Analyses of Delphi Results

Considering the objective of exploring the preferences of potential users regarding data visualization formats to be used in COVID-19 dashboards, it was deemed as relevant to understand the level of agreement regarding preferred visualisation formats. Acknowledging that a variety of methods has been used to analyse agreement within Delphi literature [41], we selected the simplest measure of agreement, the percentage of agreement for the preferred format. This is calculated as the number of times an option was chosen (frequency), divided by the total number of answers, multiplied by 100, and is a meaningful measure if nominal scales are used, which is the case of the data visualization formats [41]. Hence, the frequency and percentage of votes for each option were calculated for all the questions using the answers from rounds 1 and 2. The mode was also calculated to capture the data visualization format most voted by the participants; and to understand if a high level of agreement was obtained, a majority rule was applied, i.e., whether a data visualization format with more than 50% of votes was observed.

The degree to which a study process produces consistent results each time it is repeated is known as stability. It occurs when responses obtained in two successive rounds are shown not to be significantly different from each other, irrespective of whether a convergence of opinion occurs [42]. In this study, the percentage of participants that do not change their responses (i.e., opinion change) was taken as a measure of stability. 

As mentioned above, a summary of all comments was presented in round 2 to inform each participant of his or her position relative to the rest of the group; and participants’ comments were also analysed so as to understand their concerns, as well as whether they justified opinion change. 

### 2.5. Implications of the Delphi Results for COVID-19 Dashboard Design

In Portugal, the main official dashboard for monitoring COVID-19 data has been developed by Direção-Geral da Saúde (DGS, Directorate-General of Health), available at https://covid19estamoson.gov.pt/estado-epidemiologico-covid19-portugal/ (accessed on 2 October 2021). This platform reproduces official information regarding the number of confirmed COVID-19 cases daily, total or aggregated by a certain category—the October 2021 version is presented in Figure 3. This dashboard utilizes different data visualization formats to transmit indicators’ performance through a combination of charts, tables and maps.

Within the scope of this study, we illustrate the impact of considering the preferences of potential users in the COVID-19 dashboard configuration.

## 3. Results

### 3.1. Delphi Participation

The Web-Delphi process took place between 22 July and 14 August 2021. Each round was available for two weeks, and there was a one-week break between the end of the first and the beginning of the second round. A short time interval between rounds was established as a means of maintaining participants’ interest and reducing fatigue.

The post with an invitation to participate and complete the Web-Delphi questionnaire was shared on July 22 via the Facebook, Instagram and LinkedIn social networks. Readers were also asked to forward the link with the questionnaire to other participants. 

There were 101 participants that voluntarily answered the first round, with 72 of them (71.3% completion rate) filling their e-mail and being invited for the second round. Out of these 72 participants invited for round 2, 47 concluded the second round (65%).

The average completion time recorded for the first round was 05:55 min and 03:20 min for the second round, showing no signs of fatigue. 

Participants’ composition is shown in Table 3. It can be read that most participants that completed the Web-Delphi process are female (59.6%), young (91.5% between 20 and 29 years old) and have a high educational level (78.7% have completed a full university course). Of the participants, 29.8% had a professional activity related to health, and 38.3% were familiar with the concept of the dashboard.

### 3.2. Delphi Results

Table 4 shows the main results of round 2, portraying the mode, the correspondent percentage agreement values, as well as the most preferred data visualisation format. Full results from the Web-Delphi process are available in Appendix A.

Results indicate that line charts as the most voted data visualization format to show time-series over a 3-month period, while column charts are the most voted to visualize time-series over a 1-week period (Questions 1–6). 

The most voted data visualization formats to present COVID-19 data by socio-demographic characteristics were divided between, on the one hand, bar (column) charts and, on the other, pie charts. Bar (column) charts were the most voted data visualization to characterize COVID-19 data by age group (Questions 7 and 10) and by region (Question 9). To visualize COVID-19 data by gender (Questions 8 and 10), pie charts (alone) and bar charts (combined with age group) were the most voted formats.

Regarding map visualizations, a choropleth map (80.9%) was the preferred option to represent the number of new confirmed cases of COVID-19 per municipality since the beginning of the pandemic. Moreover, for the last question regarding the display of maps, participants preferred the map with constant infection risk within the administrative unit (80.9%).

The rate of opinion change by participants in the second round was 20%, presenting an evolution towards a higher level of agreement along the rounds, which according to the literature [42] can be interpreted as a measure of overall stability of the Delphi process. 

Participants’ comments were analysed in order to understand their concerns. In the majority of cases, the participants pointed out their preferred visualization, usually based on subjective opinion without definitive justification. However, there were 2 participants who mentioned their experience in the data visualization field and provided well-structured reasoning for selecting certain visualization formats, taking into account the indicator in question. It is noteworthy to mention evidence of a strong impact of one participant’s comment, advising not to use a pie chart with a high number of categories for the indicator “Total number of confirmed cases by age group”: a change was observed in the opinion of 6 participants, who initially chose a pie chart. 

### 3.3. Using Delphi Results for Dashboard Design

Taking into consideration the perspectives of potential users, as captured by the Delphi panel, we produced an alternative version of the DGS COVID-19 displayed in Figure 3. Using ArcGIS software [44], an adjusted version of the dashboard is presented in Figure 4. This illustrates how consultation with potential users can inform dashboard construction. It is noteworthy to mention that the information about the top 10 municipalities by cumulative incidence was omitted, as the Delphi process only covered visual information formats. 

## 4. Discussion

### 4.1. Learning about Methods

The developed approach, based on the Delphi process, was implemented to select appropriate data visualization formats for presenting information commonly presented in public web-based COVID-19 dashboards. It is noteworthy to mention that the approach integrates the most common data visualization formats that are easy to implement in any dashboard platform. Therefore, it is highly applicable to inform the construction of public health dashboards according to the preferences of potential users in a simple way.

The proposed approach contributes to dashboard literature, namely by providing a sound approach to involve a group of potential users to select the most preferred visualizations, considering the theory-based evidence. For instance, the developed approach structures the gathering of information to enlighten dashboard construction; the indicators’ data structure is taken as a starting point, afterwards mapping it to the corresponding data visualization format, also considering the communication purpose that the final visualization pretends to transmit. For that reason, the decision table was developed to select the set of most appropriate data visualization formats according to the metrics described above.

Nevertheless, the proposed approach does not provide a unique solution, and only considers respondents’ preferences. The Delphi process has been shown to ensure the usability of selected visualizations by capturing heterogeneous users’ preferences and promoting an agreement in a legitimate and dynamic way. Delphi was considered more appropriate for the dashboard design, where the heterogeneity of users implies a high number of experts and high integration and inclusion of different types of information. Therefore, in this process the pre-set of data visualization formats is proposed to the participants, and the perceptions and preferences of users are investigated by asking the preferred data visualization format for the indicator in question.

### 4.2. Interpreting Results and Their Implications

In the first round, out of 101 respondents, 72 provided their e-mail, enabling them to be invited for the second round. Specifically, the majority of participants did not fill out their e-mail at the end of the questionnaire in order to proceed with the process. This could be explained by participants’ not feeling comfortable sharing their personal information (although they were guaranteed anonymity and the use of their emails only for study purposes). 

Forty-seven participants participated in the second round of the Delphi study, with a response rate of 65%. The low response rate can be explained by low interest, an increase in fatigue between rounds, the fact that the general population are not familiar with two-round Delphi questionnaires, and eventually because the study was carried out in July and August, comprising holidays for many Portuguese.

The first part of the questionnaire explored the data visualization formats of temporal COVID-19 indicators. The obtained results suggest that a column chart is more informative to depict variations for shorter periods of time, whereas a line chart is better in displaying evolution over longer time intervals. Consequently, these results highlight the importance of using distinct visualization formats for different time intervals.

The second part of the questionnaire included questions related to the total number of confirmed cases since the beginning of the COVID-19 pandemic. The pie chart and column chart were the most voted options for this indicator aggregated by gender and region, respectively. For the indicator “Total number of confirmed cases of COVID-19 by age group”, the switch in the mode (most preferred option) between the rounds can be related to one participant’s comment, which advised not to use a pie chart with a high number of categories, which can explain the decrease of votes for that visualization format in the second round. This strong influence of the comment on results reinforces the usefulness of Delphi to promote the interaction of participants between rounds.

Regarding the map questions, participants preferred the maps with constant infection risk within the administrative unit, suggesting that the respondents avoided more complex maps with higher spatial resolution.

Considering that a group majority (as an indicator of group agreement) was established when at least 50% of the participants selected a certain data visualization format, 8 from 12 questions achieved a group agreement regarding the selection of data visualization format, and 3 out of the 4 remaining questions reached the percentage value close to 50% (46.8%). As previously mentioned, only the indicator “Total number of confirmed cases of COVID-19 by age group” showed inconclusive results and a switch in the most preferred format (i.e., mode) in the two rounds. Regarding overall results, they were considered stable, and it was possible to observe an evolution of group judgements towards a higher level of agreement along the rounds.

### 4.3. Study Limitations and Challenges

Several challenges were encountered during the development of this study. Firstly, a limited number of studies were found reflecting upon visualizations, since the research on dashboards and data visualization principles is still in its early stages, up to our knowledge with just a few recent publications. Therefore, it would be useful to develop more work to validate the proposed approach to inform dashboard design. The developed approach required the abstraction from individual graphical expressions, such as shape, colour or position, and focused on the underlying data structure, and this did not cover the aesthetics of visualizations, a topic that is worth further investigation.

The indicators’ selection stage must precede the proceedings regarding the identification of alternative visual formats, a step that requires specific thinking. One should note that the developed approach focused on data visualization selection. 

Concerning the developed Delphi process, it entails several limitations, such as: it may be influenced by participant’s bias and tendency to eliminate extreme positions to achieve central consensus [45]; it lacked empirical rules or guidelines for the definition of a level of consensus, which could be partially overcome with a workshop to discuss Delphi results; the nominal scale, utilized in this process, is simple but makes it difficult to perform additional statistical analyses; and the snowball sampling strategy is a non-random sampling method, having led to a young and educated group of respondents, for which bias should be considered in the analysis of results. To reduce possible bias sources, an effort has been made to share an invitation for a questionnaire on different social media, covering individuals with different profiles, thereby representing a wide spectrum of points of view. 

It is noteworthy to mention that this was an exploratory study that aimed to investigate methods to inform the design of data visualization tools. Given the need to intensively incorporate maps and graphs in the Web-Delphi, the platform SurveyHero was selected to execute the Delphi process. Although the platform was flexible to easily integrate multiple visual formats and it was adequate for doing a one-round questionnaire, it lacked the Delphi-related functionalities from other Web-Delphi platforms, such as Welphi [46], which for instance enables managing together all the rounds, data information and participants’ answers, and to contact participants more easily. Performing the second-round questionnaire required thus adjusting the information for each participant and sending individual emails, without automatization of the process. Accordingly, future Web-Delphi processes to inform dashboard design can be enabled by using web platforms, such as Welphi [46], to integrate multiple images as a multiple-choice option. 

## 5. Conclusions

The Web-Delphi participatory approach was a useful tool to inform the design of data visualization formats to monitor the COVID-19 pandemic in Portugal and contributes to bridging the gap between the generation of data visualization and dashboards to communicate the evolution of COVID-19 key indicators to the general public. The COVID-19 dashboard shaped from results showed how the generated knowledge can be used.

The developed methodology contributes to dashboard literature, by obtaining group knowledge about preferred data visualization formats with a transparent and replicable Web-Delphi approach. Of equal importance is the contribution to COVID-19-related literature, exploring which data formats are favoured by young and highly-educated users in Portugal. 

The proposed approach has been shown to be concise, visually effective, and efficient to promote an agreement and getting potential users’ views. It has thus potential to be more extensively used by health systems’ organizations building public health tools that should consider the views of the population. 

In order to further contribute to bridging the gap between the visualisation of information and the use of that information for policy design, further research could explore how to design and evaluate policies using visual information in COVID-19-related dashboards, for instance, exploring the use of multicriteria value modelling [30,47,48] in that context. 

## Figures and Tables

**Figure 1 ijerph-19-11012-f001:**
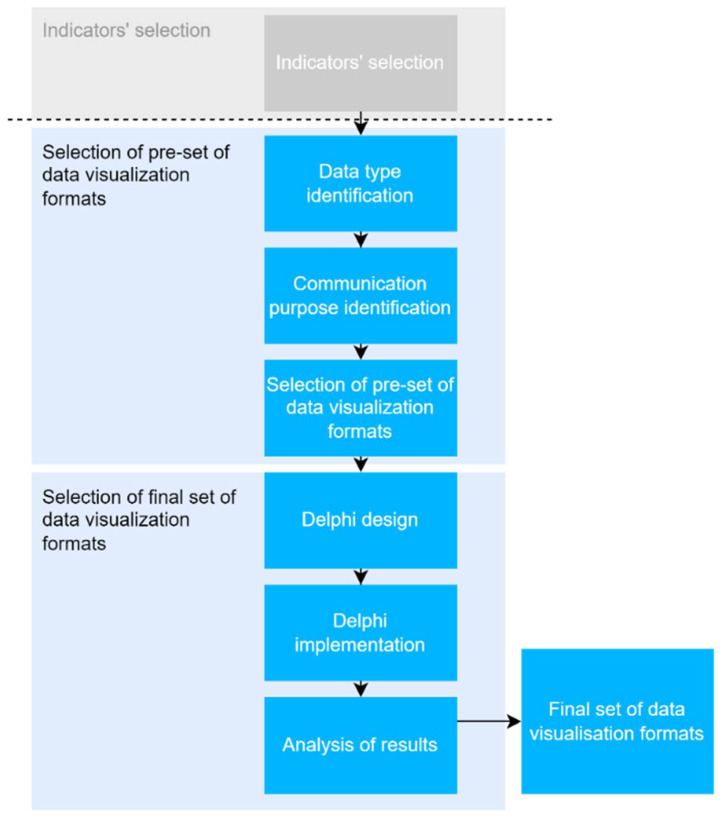
Steps performed for the implementation of the proposed approach.

**Figure 2 ijerph-19-11012-f002:**
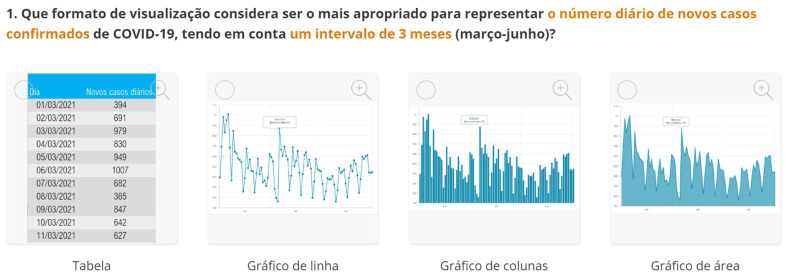
Question 1 of the first part of the Delphi questionnaire. Note: The figure content is written in Portuguese. On top of the figure, Question 1 is posed: (trans. from Portuguese) “Which visualization format do you consider to be the most appropriate to represent a daily number of new confirmed COVID-19 cases, considering three-month time interval (March–June)?”. Below the question, four visualization formats are presented and identified with the following captions (from left to right) Table; Line Chart; Column Chart; Area Chart. The header of the Table provides the number of new COVID-19 cases (right column), by day (left column). The three charts show the number of cases (*y*-axis) by day (*x*-axis).

**Figure 3 ijerph-19-11012-f003:**
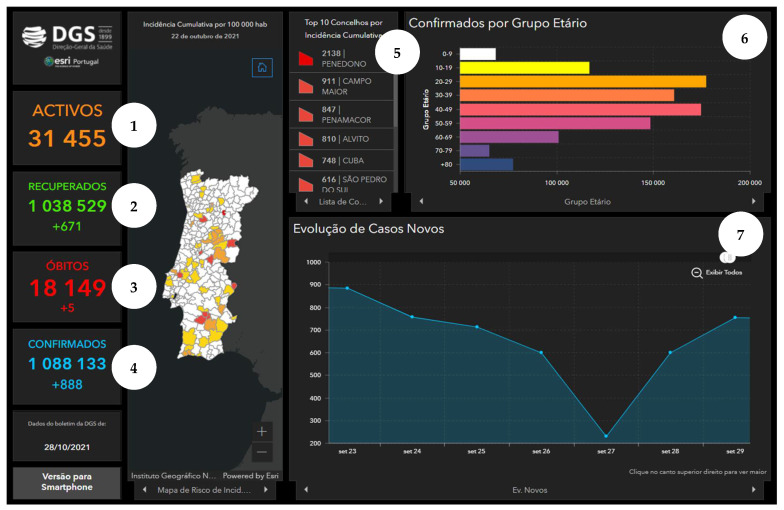
Screen of the DGS COVID-19 Dashboard available at https://covid19estamoson.gov.pt/estado-epidemiologico-covid19-portugal/, accessed on 28 October 2021. Note: The figure content is written in Portuguese. First column of the screen shows COVID-19 statistics: 1—Number of Active Cases (total), 2—Number of Recovered (total and daily), 3—Number of Deaths (total and daily), 4—Number of Confirmed Cases (total and daily); second column presents a map of Portugal with Cumulative incidence per 100,000 hab. by Municipality, on 22 October 2021; third and fourth columns show (from left to right, top to bottom): 5—List with Names and Cumulative Incidence of Top 10 Municipalities with higher incidence, 6—Number of Cases (*x*-axis) per Age Group (*y*-axis), 7—Evolution of New Cases (*y*-axis) between 23 and 29 September 2021 (*x*-axis) (screen from [43]).

**Figure 4 ijerph-19-11012-f004:**
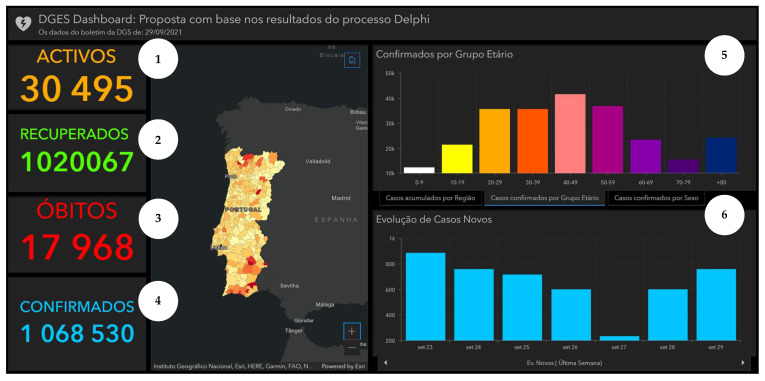
Illustrative representation of the adjusted DGS COVID-19 Dashboard produced with ArcGIS [44]. Note: The figure content is written in Portuguese. The text on top of the dashboard informs that the adjusted dashboard is a proposal based on the Web-Delphi process results. Below three columns with statistics, a map and charts are presented. First column shows COVID-19 statistics: 1—Number of Active Cases, 2—Total Number of Recovered, 3—Total Number of Deaths, 4—Number of Confirmed Cases; second column presents a map of Portugal with Cumulative incidence per 100,000 hab. by Municipality, on 29 September 2021; third column show (from top to bottom): 5—Number of Cases (*y*-axis) per Age Group (*x*-axis), 6—Evolution of New Cases between 23 and 29 September 2021). On chart 5, three tabs pointing to three charts are available (from left to right): Cumulative cases by Region, Confirmed Cases per Age Group (selected) and Confirmed Cases by Sex.

**Table 1 ijerph-19-11012-t001:** Decision table of data visualization format’s selection (CQ—1 categorical variable, 1 quantitative variable, QQ—2 quantitative variables, QQQ—3 quantitative variables, QQC—2 quantitative variables, 1 categorical variable, CCQ—2 categorical variables, 1 quantitative variable).

Variables’ Types Communication Purpose	CQ	QQ	QQQ	QQC	CCQ
**Comparing categorical values**	Bar (Column) Chart	-	-	-	Two-sided Bar Chart, Grouped Bar (Column) Chart
**Assessing hierarchies and part-to-whole relationships**	Pie Chart, Donut Chart, (100%) Stacked Bar Chart	-	-	-	(100%) Stacked Bar Chart
**Showing changes over time**	Column Chart	Line Chart, Area Chart	-	Line Chart, Area Chart	Grouped Column Chart, (100%) Stacked Column Chart
**Plotting connections and relationships**	-	Scatter Plot	Bubble Plot	-	-
**Mapping geo-spatial data**	-	-	Point map, Choropleth map, Bubble map, Dasymetric map	-	-

**Table 2 ijerph-19-11012-t002:** Indicators and corresponding data visualization formats.

Indicator	Variable Name	Attribute Type	Data Visualization Formats
**Daily number of new confirmed cases**	Number of cases	Q	Line Chart, Area Chart, Column Chart.
	Date	Q	
**Total number of confirmed cases**	Total number of cases	Q	
	Date	Q	
**Daily number of new confirmed cases per region**	Number of cases	Q	Line Chart, Area Chart, Grouped Column Chart, Stacked Column Chart.
	Date	Q	
	Region	C	
**Total number of confirmed cases by region**	Number of cases	Q	Bar Chart, Column Chart, Pie Chart, Donut Chart, 100% Stacked Bar Chart.
	Date	Q	
	Region	C	
**Total number of confirmed cases by age group**	Number of cases	Q	
	Age group	C	
**Total number of confirmed cases by gender**	Number of cases	Q	
	Gender	C	
**Total number of confirmed cases by gender and age group**	Number of cases	Q	Two-sided Bar Chart, Grouped Column Chart, Stacked Column Chart, 100% Stacked Column Chart.
	Gender	C	
	Age group	C	

**Table 3 ijerph-19-11012-t003:** Socio-demographic characteristics of first and second rounds’ participants.

		Round 1 (%)	Round 2 (%)
**Gender**			
	Male	31.9	40.4
	Female	68.1	59.6
**Age group**			
	0–19	5.6	4.3
	20–29	86.1	91.5
	30–39	1.4	2.1
	40–49	2.8	0.0
	50–59	4.2	2.1
**Highest education level**			
	Techno-professional school	6.9	4.3
	High school	13.9	17.0
	Bachelor’s degree	40.3	34.0
	Master’s degree	38.9	44.7
**Have the professional activity related to health sector**			
	Yes	31.9	29.8
	No	50.0	46.8
	Do not exercise any professional activity	18.1	23.4
**Are familiar with COVID-19 dashboards**			
	Yes	47.2	38.3
	No	52.8	61.7

**Table 4 ijerph-19-11012-t004:** Main Web-Delphi results for the second round (majority percentages in bold). Note: Some figure contents are written in Portuguese. On Indicators 1–6, data labels (callouts) inform the value of the indicator pointing to a specific day; On Indicators 8 and 10, “Homens” refers to Males and “Mulheres” to Females; The legend title in Indicator 11 refers to “Number of new confirmed cases”.

	Indicator	Mode	Percentage Agreement	Most Preferred Data Visualization Format
1	Daily number of new confirmed COVID-19 cases (3-month interval)	Line Chart	46.8%	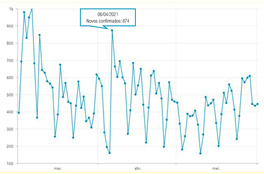
2	Daily number of new confirmed COVID-19 cases (1-week interval)	Column Chart	46.8%	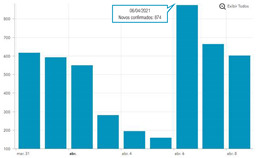
3	Total number of confirmed COVID-19 cases (3-month interval)	Line Chart	**63.8%**	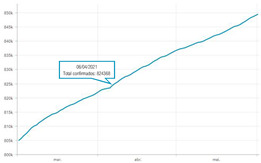
4	Total number of confirmed COVID-19 cases (1-week interval)	Column Chart	**61.7%**	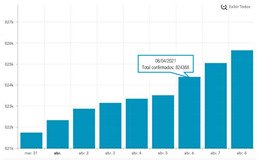
5	Daily number of new confirmed COVID-19 cases for three regions—Norte, Centro e Alentejo (3-month interval)	Line Chart	46.8%	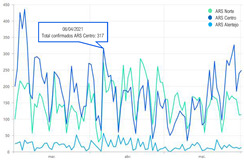
6	Daily number of new confirmed COVID-19 cases for three regions—Norte, Centro e Alentejo (1-week interval)	Column Chart	**53.2%**	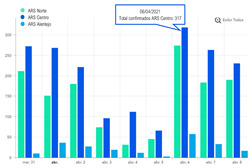
7	Total number of confirmed COVID-19 cases per age group	Column Chart	40.4%	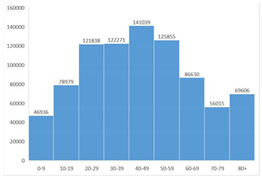
8	Total number of confirmed COVID-19 cases per gender	Pie Chart	**74.5%**	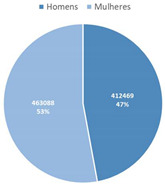
9	Total number of confirmed COVID-19 cases per region	Column Chart	**51.1%**	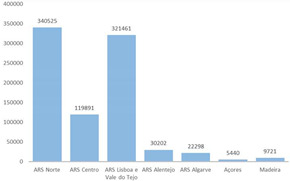
10	Total number of confirmed COVID-19 cases per gender and age group	Two-sided Bar Chart	**78.7%**	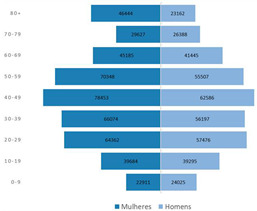
11	Number of new confirmed COVID-19 cases per municipality since the beginning of pandemic (incidence)	Choropleth map	**80.9%**	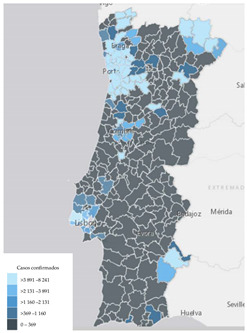
12	Number of new confirmed COVID-19 cases per 100 k inhabitants, per municipality since the beginning of pandemic (cumulative incidence)	A risk is constant within administrative limit	**80.9%**	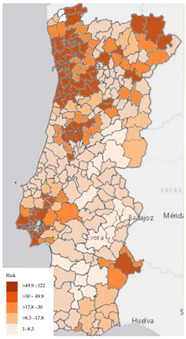

## Data Availability

The Web-Delphi questionnaire data is printed in Appendix A. A file with the Web-Delphi questionnaire data can be obtained upon request.

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
