# Peer review of "Informing the Design of Data Visualization Tools to Monitor the COVID-19 Pandemic in Portugal: A Web-Delphi Participatory Approach"

_ijerph, 2022, doi:10.3390/ijerph191711012_

Round 1

Reviewer 1 Report

This is a well written paper clearly contributing to the field researching and developing data visualization. The use of the Web-Delphi participatory approach is well documented, and gave valuable results.

The only observation of this peer about the research is a critique about the method to select the respondents via social media by a snowball approach. But authors already stated that this approach is questionable, and it doesn't affect the values of the Delphi method applied to such dashboard design processes.

Applying this methodology to a dashboard monitoring the COVID-19 pandemic was a valid choice. But will this choice make this valuable research suitable for publication in the International Journal of Environmental Research and Public Health? No health related or environment related findings were made, this is a paper from a completely different discipline.

This peer had to make a hard choice. I could accept the work in its present form in any journal relating to data science, web design, or even participatory methodologies. But this paper, though using the covid dashboard, is not related in any sense to the topics of IJERPH. I advise the authors to submit this excellent paper to a related journal!

Reviewer 2 Report

A custom-built Web-Delphi approach was developed to investigate the general public's preferences and viewpoints on various data visualization formats, with the goal of determining which visualization formats should be featured inside dashboards for the COVID-19 pandemic. This article addresses the public's desire for improved visualization of a COVID-19 platform. However, the article's structure needs some work before it can be published.

1. Abstract: The abstract of the article provides far too much background context and far too little information about the article's findings. What aspects were the subjects more satisfied with, and what were the authors' main findings?

2. Method: Rather than the results, the methods section should cover how participants were recruited and demographic information about the participants.

3. Conclusion: This section has too much content that needs to be woken up and refined. What is the purpose of the article? Reiterating the limitations would undermine the article's contribution and scientific significance.

Reviewer 3 Report

Dear Authors

With the usual greetings, I pray that you are in good health. Regarding the paper, I congratulate you on the study proposal, which deals with a current topic and proposes an opportunity for improvement in how health managers visualize the data that help in decision making. Regarding the structure of the paper, I will indicate the positive points:

1. The Abstract fulfilled its objective, providing the reader with the study's goals, methods, results, and limitations.

2. The introduction brings the study context and briefly presents the scope, methodology, and results; I consider the introduction well structured.

3. The methodology was very well detailed.

4. References are current.

As a point of improvement, I can indicate that:

1. Figures 1-4 does not allow proper visualization of information.

2. In the same way as item 1, the graphics in table 4 do not allow a proper visualization.

I believe that after these improvements, the paper will be ready for publication.

Best Regards

Round 2

Reviewer 1 Report

I can see the effort made to fit this study with public health issues. If the editors think it fits into their journal, I have no more objections, though I retain my opinion that this paper is less related to public health issues than to data visualization and other fields.